# Adverse events following COVID-19 vaccination: A comprehensive analysis of spontaneous reporting data in Ghana

Amma Frempomaa Asare[1], George Tsey Sabblah[1‡], Richard Osei Buabeng[1‡], Yakubu Alhassan[2☉], Abena Asamoa-Amoakohene[1‡], Kwame Amponsa-Achiano[3‡], Naziru Tanko Mohammed[3‡], Delese Mimi Darko[1‡], Harriet Affran Bonful[4☉*]

1 Food and Drugs Authority, Accra, Ghana, 2 Department of Biostatistics, School of Public Health, University of Ghana, Legon, Accra, Ghana, 3 Ghana Health Service, Expanded Programme on Immunization, Accra, Ghana, 4 Department of Epidemiology and Disease Control, School of Public Health, University of Ghana, Legon, Accra, Ghana

☉ These authors contributed equally to this work.
‡ GTS, ROB, AAA, KAA, NTM and DMD also contributed equally to this work.
* habonful@ug.edu.gh

**Data Availability Statement:** The data obtained from the Ghana FDA for use in this work has been published on Mendeley repository and can be

## Abstract

Vaccines are important public health tools and formed part of the fight against the COVID-19 pandemic. Five COVID-19 vaccines were given Emergency Use Authorization in Ghana and deployed during the pandemic. Early phase trials of the vaccines were mostly not conducted in Africans. This study examines safety data during their deployment under real-life conditions in Ghana. This study analysed secondary data on COVID-19 vaccine-related adverse events following immunization (AEFI) reported to the Ghana Food and Drugs Authority (GFDA) between March 2021 and June 2022 using STATA. AEFIs were coded with their Preferred Terms using the Medical Dictionary for Regulatory Activities, version 24.0. Statistical tests examined associations between demographic characteristics, vaccine types, seriousness, and AEFI outcomes. Binary logistic regression model assessed factors associated with serious AEFIs, while the GFDA's Joint COVID-19 Vaccine Safety Review Committee provided causality assessments of serious AEFIs. Overall cumulative incidence of AEFIs was about 25 per 100,000 persons vaccinated. Across the five vaccines, majority of the AEFIs reported were not serious (98.7%) with higher incidences in those below 50 years (74.0%) and females (51.2%). The most common AEFIs recorded were headache (52.9%), pains (44.4%), pyrexia (35.1%), chills (16.7%) and injection site pain (15.6%). Relative to those 50 years and above, the odds of serious AEFI were 60% less among those aged <30 years (aOR = 0.40, CI: [0.19, 0.86], p = 0.019). However, a causality assessment of the 57 serious AEFIs indicated only 8 (14%) were vaccine product-related. There was a low incidence of AEFIs following deployment of the vaccines in Ghana with a much lower incidence of serious AEFIs. Informing the public about the safety of the vaccines and potential side effects may increase trust and acceptance, decreasing hesitancy in current and future vaccination programmes.

accessed at https://data.mendeley.com/datasets/kfxc7xv4ym/1.

**Funding:** The authors received no specific funding for this work.

**Competing interests:** The authors have declared that no competing interests exist.

## Introduction

Vaccines are widely regarded as one of medicine's most impressive success stories in global health. They have significantly reduced targeted infectious diseases in all documented cases, playing an outstanding role in reducing human disease morbidity and mortality [1]. This makes vaccination the primary control method against infectious disease transmission. Unfortunately, just like other drugs, vaccines are associated with adverse reactions or events commonly referred to as adverse events following immunization or simply, AEFIs [2]. An AEFI is any untoward medical occurrence which happens after vaccination and does not necessarily have a causal relationship with the use of the vaccine [3, 4]. Adverse reactions to vaccines may occur because of the vaccine recipient's intrinsic properties such as age, sex, race/ethnicity, weight, and pre-existing medical condition [5, 6]. Additionally, the vaccine's administration and composition parameters, such as the method of administration or the location, might affect the vaccine's safety profile in some individuals [7].

The global endeavour to battle the COVID-19 pandemic was characterised by the rapid development and implementation of vaccinations, which was unprecedented [8]. Prior to the emergence of coronaviruses, no vaccines had been developed for humans against these viruses [9]. The swift pace of the development of the COVID-19 vaccines as well as the inadequate time for follow-up after vaccination resulted in great public anxiety about the safety of the vaccines [10]. This is especially so since new platforms such as ribonucleic acid (RNA) were used for the development of some of these vaccines [11]. Results from a research by Dodd et al. indicated that some persons were sceptical about receiving the vaccines with common reasons being the safety and possible adverse reactions to these vaccines [12]. These findings were found to be consistent with a cross-sectional online survey conducted among Ghanaian adults [13], before the rollout of the vaccines in Ghana on 2nd March 2021.

As countries work towards achieving extensive vaccination coverage with COVID-19 vaccines, a vital part of this significant endeavour is the ongoing monitoring and evaluation of vaccine safety. AEFIs are an important part of monitoring the safety of COVID-19 vaccination as they offer valuable information on the actual effects of the vaccine in real-world scenarios.

Clinical trials and post-approval safety data available on approved COVID-19 vaccines found them to be generally safe with adverse reactions usually non-serious, transient, and self-limiting [14]. COVID-19 vaccine-related adverse drug reactions have been reported in over 80% of vaccine recipients in studies conducted in many countries across the globe such as the Czech Republic, Iran, Jordan, and Iraq. Some of the commonly reported adverse drug reactions include injection site pain, redness, tiredness, or body ache [5, 6, 15, 16].

Although most of the AEFIs reported to be associated with the COVID-19 vaccines are generally mild to moderate, there have been reports of some very rare and serious reactions including deaths associated with the use of the vaccines. For instance, the mRNA vaccines like Pfizer-BioNTech and Moderna have been linked to Guillain-Barré Syndrome (GBS), myocarditis and pericarditis [17–19]. In addition, reports of thrombosis with thrombocytopenia syndrome (TTS) have also been recorded [19, 20]. The adenovirus viral vector-based vaccines like Johnson and Johnson and Covishield (AstraZeneca) have also been linked to TTS and pulmonary embolism [21–23].

In Ghana, five vaccines were given Emergency Use Authorization (EUA) as of 20th December 2021 by the National Regulatory Agency, the Food and Drugs Authority (FDA) for deployment in the mass vaccination programme following the COVID-19 outbreak. These vaccines were AstraZeneca, Moderna, Pfizer-BioNTech, Johnson and Johnson (J&J) and Sputnik V [24, 25]. As of 30th June 2022, the total number of COVID-19 vaccine doses administered in Ghana (AstraZeneca, Moderna, Pfizer, Johnson and Johnson and Sputnik V) was 17,409,005 [26].

The number of persons who had taken at least 1 dose of any of the five COVID-19 vaccines was 10,733,719, representing 47% of the targeted 22.9 million persons (33.8% of the entire Ghanaian population). In addition, persons fully vaccinated (received 2 doses of AstraZeneca, Pfizer, Moderna and Sputnik V or received one dose of J&J) were 7,510,586 which was 32.9% of the targeted 22.9 million (23.7% of the total Ghanaian population). Finally, the number of persons receiving their 1st booster dose was 1,192,595 [26].

The AEFIs which were reported and required treatment were treated in line with the WHO guidelines on case definitions and treatments for AEFIs [27]. However, most of the non-serious AEFIs were self-limiting and resolved on their own. To control the spread of COVID-19 disease, the approved vaccines need to be more widely used [28]. It is therefore critical for both the vaccine recipient and the healthcare professional who delivers vaccinations to be aware of the potential adverse drug reactions associated with these vaccines. If a vaccine is deemed to be overly reactogenic, it may lead to a person refusing subsequent doses leading to poor vaccination rates and consequent under-protection of the community. AEFIs can be context-specific (residence, race, or ethnicity) [15, 29] and although COVID-19 AEFIs have been reported in other parts of the world, the case of Ghana might be different. This is especially so since most of the safety data gathered during the early development phase of these vaccines were in populations typically of non-African descent such as the USA, Asia, and Europe [2, 30, 31]. It is necessary to assess the safety of these vaccines under real-life conditions (post-approval) among Ghanaians and determine the vaccine recipients most vulnerable to adverse events. This will better inform the planning of future vaccination programmes. Furthermore, informing the public about potential side effects to expect after COVID-19 vaccinations, may increase public trust and acceptance (decrease vaccine hesitancy) in current and future vaccination programmes.

This manuscript presents a comprehensive review of spontaneous reporting data on adverse events following COVID-19 vaccination in Ghana and reports of the FDA's Joint COVID-19 Vaccine Safety Review. The primary objective is to contribute to the evolving understanding of vaccine safety within the unique context of a low and middle-income country. Ghana, like many other countries, has actively participated in global vaccination campaigns, and an in-depth analysis of AEFI data specific to this region is paramount for tailoring public health responses and optimizing vaccine safety strategies.

## Materials and methods

### Study design

This was an analytic cross-sectional study involving secondary data analyses of COVID-19 vaccine-related AEFI reports in the safety database of the Ghana FDA, followed by analysis of causality assessment reports on all serious AEFIs. These reports in the database were a collection of AEFI received through the passive or spontaneous reporting system using various reporting tools; phone calls, paper reporting forms, mobile application (Med Safety App) and online reports and covered the period of 2nd March 2021 to 30th June 2022. The date of 2nd March 2021 was when the mass vaccination campaign began in Ghana and 30th June 2022 was chosen as the cut-off date since approval to the request for use of the FDA's safety data was given in June 2022. Causality assessment reports on all serious AEFIs which occurred over the same period were obtained from the FDA's Joint COVID-19 Vaccine Safety Review Committee [32]. The Committee in its assessment, employed the procedure outlined in the World Health Organization (WHO) User Manual for the Revised WHO Classification on Causality Assessment of An Adverse Event Following Immunization (AEFI) [33].

## Study population

The study involved COVID-19 vaccine AEFI data which were obtained through spontaneous reporting by individuals aged 18 years or older in Ghana who had received at least one dose of any of the five vaccines deployed in the nationwide mass vaccination campaign from 2nd March 2021 to 30th June 2022. As of 30th June 2022, 17,409,005 doses had been administered in Ghana as follows; AtraZeneca, 10,096,925; Sputnik-V, 17,982; Moderna, 1,065,357; Pfizer-BioNTech, 3,910,669 and Johnson and Johnson, 2,318,072 [26].

## Eligibility criteria

This study included AEFIs in the FDA's safety database received from healthcare professionals and vaccine recipients aged 18 years or older who had received at least one dose of any of the five COVID-19 vaccines deployed during the pandemic. Reports from the active safety surveillance were excluded.

## Study variables

**Dependent or outcome variables.** The outcome variables included the type, time to onset of AEFIs (latency period of AEFI), the seriousness of the AEFIs experienced, and the outcome of the AEFIs, which were defined as recovery, not yet recovered, death or unknown. AEFIs were graded as serious and non-serious per the WHO standard definition, which defines a serious event as "any untoward medical occurrence that results in death, hospitalization or prolongation of hospitalization, persistent or significant disability/incapacity, results in congenital anomaly or is life-threatening" [33].

**Independent variables.** Vaccine recipients' age, sex, co-morbidities, as well as the vaccine dose and type of vaccine received, were considered independent variables, and these were examined in relation to the outcome variables to understand their potential associations.

## Data collection and handling/processing

A password-protected laptop was used to store the de-identified AEFI data obtained from the FDA's safety database on 20th December 2022. Additionally, the data on the causality assessment of serious AEFIs which was carried out by the Joint COVID-19 Vaccine Safety Review Committee was obtained on 11th December 2023. A total of 4,295 individuals were identified by their unique codes. After receiving the data, the data was cleansed to remove unwanted observations and structural errors were corrected. The AEFI data obtained were coded with their Preferred Terms using the Medical Dictionary for Regulatory Activities (MedDRA version 24.0) [34].

## Statistical analysis

Stata MP version 18 (StataCorp, College Station, TX, USA) was used to analyse the data. Descriptive characteristics of vaccine recipients were done using frequency and percentages for categorical variables, and median and interquartile range for continuous variables. Bar and pie charts were also used to describe selected characteristics such as chronic medical conditions and outcomes of AEFIs.

Descriptive analysis was performed across the various vaccine types. The Pearson chi-square test was used to assess the association between the background characteristics and the type of vaccines, the seriousness of AEFIs and the outcome of the AEFIs. Where appropriate, the Fisher's exact test was used instead of the Pearson chi-square test. The Wilcoxon rank sum

and the Kruskal Wallis test were used to test the equality of medians between two and three or more groups respectively.

The penalized binary logistic regression model was used to assess the factors associated with the seriousness of AEFIs and AEFIs with death as an outcome. The user-written command *"firthlogit"* with the option of reporting odds ratios instead of coefficients was used to estimate the odds ratios. The penalized model was used due to the low percentage of vaccine recipients who experienced serious AEFI in general and those with death as an outcome. All statistical analyses were considered significant at an alpha of ≤0.05 level.

## Ethical considerations

Ethical approval for the study was granted by the Ghana Health Service Ethics Review Committee (GHS-ERC) with reference number GHS-ERC: 051/09/22 dated 18[th] October 2022. Permission was also granted by the FDA for the use of the data. To ensure the confidentiality of vaccine recipients, the electronic AEFI data were de-identified and the data was kept safe from unauthorized access through encrypted files on a password-protected laptop while preventing accidental loss or destruction.

## Results

### Descriptive characteristics of the spontaneous report

A total of 8,498 AEFI reports were obtained from the FDA's COVID-19 safety database, out of which 4,295 were identified as having been spontaneously reported by vaccine recipients and healthcare professionals. The other AEFIs had been reported through active surveillance but were excluded from the analyses as this was not the study's focus.

A total of 3,590 of the AEFI reports received through the spontaneous system (83.6%) were from AstraZeneca, with 524 (12.2%) recorded from Sputnik V, 63 (1.5%) from Johnson and Johnson, 61 from (1.4%) Moderna vaccine and 57 (1.3%) from Pfizer. The median age of the vaccine recipients was 33 years (IQR: 28–41 years). Most of the reported cases were within the age group of 30–49 years (43.3%). More than half were males (51.2%). Of those who reported AEFIs, 92.2% had received only the first dose with 5.0% receiving their second dose. A total of 293 (6.8%) reported having an existing medical condition; hypertension (49.5%), diabetes (12.6%), stomach ulcer (11.6%) and asthma (10.6%) were the most common. Table 1 describes the characteristics of the individuals reporting AEFIs through the spontaneous surveillance system by vaccine type.

### Types of AEFIs reported in Ghana

The most common AEFIs reported were headache (52.9%), pains (44.4%), pyrexia (35.1%), chills (16.7%) and injection site pain (15.6%) (Fig 1).

Table 2 shows the AEFIs experienced by vaccine type among the vaccine recipients.

Table 3 shows the Cumulative Incidence of AEFIs from the spontaneous reports per the total number of vaccine doses administered among Ghanaians by vaccine type. A cumulative incidence of 24.7 per 100,000 persons vaccinated was observed. The highest cumulative incidence of 2,914 per 100,000 persons vaccinated was observed among Sputnik V vaccine recipients, while recipients of the Pfizer vaccine reported the least cumulative incidence of 1.5 per 100,000 (Table 3).

**Table 1. Descriptive characteristics of individuals who reported AEFIs following COVID-19 vaccination in Ghana.**

| Characteristics | | Vaccine type | | | | | P-value |
|---|---|---|---|---|---|---|---|
| | **Total** | **AstraZeneca** | **J&J** | **Moderna** | **Pfizer** | **Sputnik V** | |
| | **N = 4,295** | **N = 3,590** | **N = 63** | **N = 61** | **N = 57** | **N = 524** | |
| | **n (%[c])** | **n (%[c])** | **n (%[c])** | **n (%[c])** | **n (%[c])** | **n (%[c])** | |
| **Age in years, median (IQR)** | 33 (28–41) | 33 (28–42) | 32 (22–42) | 27 (21–35) | 32 (23–41) | 31 (27–38) | <0.001 [k] |
| **Age group** | | | | | | | <0.001 |
| <30 years | 1,317 (30.7) | 1,054 (29.4) | 20 (31.7) | 28 (45.9) | 21 (36.8) | 194 (37.0) | |
| 30–49 years | 1,860 (43.3) | 1,572 (43.8) | 22 (34.9) | 8 (13.1) | 13 (22.8) | 245 (46.8) | |
| 50+ years | 578 (13.5) | 510 (14.2) | 4 (6.3) | 6 (9.8) | 11 (19.3) | 47 (9.0) | |
| Not specified | 540 (12.6) | 454 (12.6) | 17 (27.0) | 19 (31.1) | 12 (21.1) | 38 (7.3) | |
| **Sex** | | | | | | | <0.001 |
| Female | 2,197 (51.2) | 1,894 (52.8) | 26 (41.3) | 39 (63.9) | 31 (54.4) | 207 (39.5) | |
| Male | 2,087 (48.6) | 1,689 (47.0) | 37 (58.7) | 22 (36.1) | 26 (45.6) | 313 (59.7) | |
| Not stated | 11 (0.3) | 7 (0.2) | 0 (0.0) | 0 (0.0) | 0 (0.0) | 4 (0.8) | |
| **Dose vaccine** | | | | | | | <0.001 |
| 1st dosage | 3,962 (92.2) | 3,333 (92.8) | 63 (100.0) | 37 (60.7) | 27 (47.4) | 502 (95.8) | |
| 2nd dosage | 213 (5.0) | 201 (5.6) | 0 (0.0)* | 6 (9.8) | 6 (10.5) | 0 (0.0) | |
| Not specified | 120 (2.8) | 56 (1.6) | 0 (0.0) | 18 (29.5) | 24 (42.1) | 22 (4.2) | |
| **Have chronic condition** | | | | | | | <0.001 [f] |
| No | 4,002 (93.2) | 3,299 (91.9) | 63 (100.0) | 60 (98.4) | 57 (100.0) | 523 (99.8) | |
| Yes | 293 (6.8) | 291 (8.1) | 0 (0.0) | 1 (1.6) | 0 (0.0) | 1 (0.2) | |
| **Medical condition (N = 293):** | | | | | | | |
| Hypertension | 145 (49.5) | 145 (49.8) | - | 0 (0.0) | - | 0 (0.0) | 1.00 [f] |
| Diabetes | 37 (12.6) | 37 (12.7) | - | 0 (0.0) | - | 0 (0.0) | 1.00 [f] |
| Ulcer | 34 (11.6) | 33 (11.4) | - | 0 (0.0) | - | 1 (100.0) | 0.21 [f] |
| Asthma | 31 (10.6) | 31 (10.7) | - | 0 (0.0) | - | 0 (0.0) | 1.00 [f] |
| Allergies | 10 (3.4) | 10 (3.4) | - | 0 (0.0) | - | 0 (0.0) | 1.00 [f] |
| Sickle cell disease | 9 (3.1) | 9 (3.1) | - | 0 (0.0) | - | 0 (0.0) | 1.00 [f] |
| Hepatitis B | 6 (2.0) | 6 (2.1) | - | 0 (0.0) | - | 0 (0.0) | 1.00 [f] |
| Abnormal cholesterol level | 6 (2.0) | 6 (2.1) | - | 0 (0.0) | - | 0 (0.0) | 1.00 [f] |
| Cancer | 3 (1.0) | 3 (1.0) | - | 0 (0.0) | - | 0 (0.0) | 1.00 [f] |
| Gastritis | 2 (0.7) | 2 (0.7) | - | 0 (0.0) | - | 0 (0.0) | 1.00 [f] |
| Typhoid | 1 (0.3) | 1 (0.3) | - | 0 (0.0) | - | 0 (0.0) | 1.00 [f] |
| UTI | 1 (0.3) | 1 (0.3) | - | 0 (0.0) | - | 0 (0.0) | 1.00 [f] |
| Other medical conditions** | 46 (15.7) | 45 (15.5) | - | 1 (100.0) | - | 0 (0.0) | 0.29 [f] |

*The 0% recorded for the 2nd dose of the J&J COVID-19 vaccine is because the vaccine is given as a single dose in the primary vaccination series.

**Other medical conditions inlcude: Eye problems, rashes, rheumatism, prostate enlargement, anaemia, diarrhoea etc.

## Number of AEFIs experienced after receiving COVID-19 vaccine dose

Table 4 shows the number of AEFIs experienced by the various observed characteristics. The median number of AEFIs experienced was 2 (IQR: 1–3 AEFIs).

Majority of those who took Sputnik reported only 1 AEFI (58.0%). Multiple symptoms were reported among more than half of those who took AstraZeneca (76.6%), J&J (65.1%), Moderna (67.2%) and Pfizer (54.4%). There were significant variations between the number of symptoms reported and age group (p<0.001), sex (p<0.001), vaccine type (p<0.001) and the dose of vaccine received (p<0.001) (Table 4).

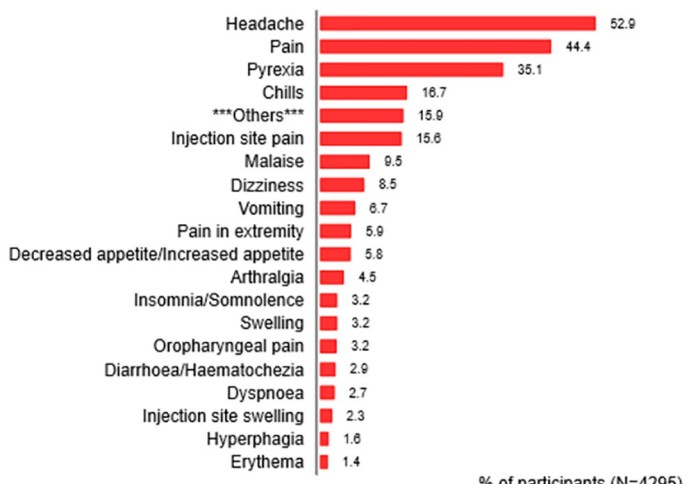

**Fig 1. Types of AEFIs reported.**

**Table 2. Types of AEFIs reported by vaccine type in Ghana.**

| Characteristics | Total | Type of vaccine | | | | | Fishers exact |
|---|---|---|---|---|---|---|---|
| | | AstraZeneca | Johnson and Johnson | Moderna | Pfizer | Sputnik V | |
| | N = 4,295 | N = 3,590 | N = 63 | N = 61 | N = 57 | N = 524 | p-value |
| Headache | 2,274 (52.9) | 2,050 (57.1) | 23 (36.5) | 17 (27.9) | 12 (21.1) | 172 (32.8) | <0.001 |
| Pain | 1,906 (44.4) | 1,696 (47.2) | 23 (36.5) | 26 (42.6) | 16 (28.1) | 145 (27.7) | <0.001 |
| Pyrexia | 1,509 (35.1) | 1,404 (39.1) | 16 (25.4) | 11 (18.0) | 7 (12.3) | 71 (13.5) | <0.001 |
| Chills | 716 (16.7) | 635 (17.7) | 10 (15.9) | 6 (9.8) | 5 (8.8) | 60 (11.5) | 0.001 |
| Injection site pain | 671 (15.6) | 452 (12.6) | 9 (14.3) | 13 (21.3) | 4 (7.0) | 193 (36.8) | <0.001 |
| Malaise | 408 (9.5) | 399 (11.1) | 1 (1.6) | 1 (1.6) | 1 (1.8) | 6 (1.1) | <0.001 |
| Dizziness | 363 (8.5) | 316 (8.8) | 1 (1.6) | 2 (3.3) | 4 (7.0) | 40 (7.6) | 0.13 |
| Vomiting | 288 (6.7) | 268 (7.5) | 3 (4.8) | 3 (4.9) | 6 (10.5) | 8 (1.5) | <0.001 |
| Pain in extremity | 255 (5.9) | 215 (6.0) | 6 (9.5) | 12 (19.7) | 8 (14.0) | 14 (2.7) | <0.001 |
| Decreased appetite/Increased appetite | 249 (5.8) | 222 (6.2) | 1 (1.6) | 4 (6.6) | 1 (1.8) | 21 (4.0) | 0.096 |
| Arthralgia | 195 (4.5) | 179 (5.0) | 2 (3.2) | 0 (0.0) | 2 (3.5) | 12 (2.3) | 0.026 |
| Swelling | 139 (3.2) | 121 (3.4) | 5 (7.9) | 6 (9.8) | 6 (10.5) | 1 (0.2) | <0.001 |
| Insomnia/Somnolence | 139 (3.2) | 119 (3.3) | 3 (4.8) | 1 (1.6) | 1 (1.8) | 15 (2.9) | 0.8 |
| Oropharyngeal pain | 136 (3.2) | 128 (3.6) | 1 (1.6) | 0 (0.0) | 0 (0.0) | 7 (1.3) | 0.018 |
| Diarrhea/Hematochezia | 126 (2.9) | 110 (3.1) | 2 (3.2) | 2 (3.3) | 5 (8.8) | 7 (1.3) | 0.019 |
| Dyspnea | 118 (2.7) | 103 (2.9) | 4 (6.3) | 4 (6.6) | 6 (10.5) | 1 (0.2) | <0.001 |
| Injection site swelling | 98 (2.3) | 86 (2.4) | 2 (3.2) | 5 (8.2) | 1 (1.8) | 4 (0.8) | 0.004 |
| Hyperphagia | 68 (1.6) | 62 (1.7) | 0 (0.0) | 2 (3.3) | 1 (1.8) | 3 (0.6) | 0.19 |
| Erythema | 61 (1.4) | 55 (1.5) | 1 (1.6) | 1 (1.6) | 0 (0.0) | 4 (0.8) | 0.59 |
| ***Other AEFIs*** | 685 (15.9) | 599 (16.7) | 18 (28.6) | 11 (18.0) | 13 (22.8) | 44 (8.4) | <0.001 |

*** Combination of all AEFIs with overall prevalence <1.0%. This includes Catarrh, ageusia, burning sensation in the body, loss of consciousness, erythema, hypoacusis, dyspnoea etc.

**Table 3. Cumulative incidence of AEFIs from the spontaneous reports.**

| Vaccine | Total number of Doses Given | Number of Spontaneous AEFI Reports | Incidence per 100,000 persons |
|---|---|---|---|
| AstraZeneca | 10,096,925 | 3,590 | 35.55537948 |
| Moderna | 1,065,357 | 61 | 5.725780184 |
| Pfizer | 3,910,669 | 57 | 1.457551125 |
| Sputnik | 17,982 | 524 | 2,914.025136 |
| Janssen | 2,318,072 | 63 | 2.717775807 |
| Total | 17,409,005 | 4,295 | 24.67114002 |

## Seriousness of AEFIs

Among the 4,295 COVID-19 vaccine recipients who reported AEFIs, 1.3% (95% CI:1.0%–1.7%) reported serious AEFIs. Serious AEFIs reported was 1.0% (CI: 0.7–1.3%) among those who received AstraZeneca, 11.1% (CI: 5.4–21.5%) among those who received J&J vaccine, 6.6% (CI: 2.5–16.2%) among those who received Moderna, 17.5% (CI: 9.7–29.6%) among those who received Pfizer. None of those who took the Sputnik V vaccine reported a serious AEFI. The seriousness of AEFI significantly varied by the type of vaccine taken (p<0.001) (Fig 2).

## Factors associated with serious AEFIs among those who reported AEFIs

Table 5 shows the incidence of serious AEFIs among those who reported AEFIs following COVID-19 vaccination by the various characteristics of the vaccine recipients. From the bivariate chi-square test, age group (p = 0.002), sex (p = 0.041), vaccine type (p<0.001) and number of vaccine dosage taken (p<0.001) were significantly associated with reported serious AEFIs.

Table 5 also shows the crude and adjusted odds ratio of serious AEFIs across the observed characteristics of the vaccine recipients. From the adjusted model, compared to those aged 50 years and above, the adjusted odds of serious AEFIs were about 60% less among those aged below 30 years (aOR: 0.40, 95% CI: 0.19–0.86, p = 0.019). Also, relative to those who took J&J vaccine, the adjusted odds of serious AEFIs was 95% less among those who took AstraZeneca vaccine (aOR: 0.05, 95% CI: 0.02-0.l2, p<0.001), 85% less among those who took Moderna (aOR: 0.15, 95% CI: 0.03–0.63, p = 0.010) and less 99% among those who took Sputnik V (aOR<0.01, 95% CI: <0.01–0.07, p<0.001).

## Causality assessment of serious AEFIs

A total of 57 AEFIs were classified as serious per the WHO's definition of 'serious AEFI'. A causality assessment was carried out on these serious AEFIs by the FDA's Joint COVID-19 Vaccine Safety Review Committee using the procedure outlined in the World Health Organization (WHO) User Manual for the Revised WHO Classification on Causality Assessment of An Adverse Event Following Immunization (AEFI) [33]. Of the 57 cases, 26 cases were rated as coincidental, 17 were ineligible for assessment, and 8 were vaccine product-related reactions. In addition, 2 events were rated as immunization error-related, 2 cases of immunization anxiety-related reaction with 2 cases having a temporal relationship but there was insufficient definitive evidence for classifying these as vaccine-causing events (Fig 3).

## Vaccine product-related serious AEFIs

Eight (8) serious AEFIs were assessed by the FDA's Joint COVID-19 Vaccine Safety Review Committee to be causally related to the vaccines. Of these, 5 were diagnosed as febrile illness

**Table 4. Number of AEFIs experienced after receiving COVID-19 vaccine dose by characteristics.**

| Characteristics | Total | Number of AEFIs experienced | | | | | | | | | Median (IQR) | P-value |
|---|---|---|---|---|---|---|---|---|---|---|---|---|
| | | 1 | 2 | 3 | 4 | 5 | 6 | 7 | 8+ | P-value | | |
| | | n (%ʳ) | n (%ʳ) | n (%ʳ) | n (%ʳ) | n (%ʳ) | n (%ʳ) | n (%ʳ) | n (%ʳ) | | | |
| **Overall** | **4,295** | **1,213 (28.2)** | **1,227 (28.6)** | **1,015 (23.6)** | **493 (11.5)** | **219 (5.1)** | **89 (2.1)** | **30 (0.7)** | **9 (0.2)** | | **2 (1, 3)** | |
| **Age group** | | | | | | | | | | <0.001 | | <0.001 ᴷ |
| <30 years | 1,317 | 376 (28.5) | 343 (26.0) | 309 (23.5) | 172 (13.1) | 74 (5.6) | 28 (2.1) | 12 (0.9) | 3 (0.2) | | 2 (1, 3) | |
| 30–49 years | 1,860 | 483 (26.0) | 560 (30.1) | 450 (24.2) | 200 (10.8) | 105 (5.6) | 47 (2.5) | 9 (0.5) | 6 (0.3) | | 2 (1, 3) | |
| 50+ years | 578 | 210 (36.3) | 173 (29.9) | 115 (19.9) | 41 (7.1) | 25 (4.3) | 7 (1.2) | 7 (1.2) | 0 (0.0) | | 2 (1, 3) | |
| Not specified | 540 | 144 (26.7) | 151 (28.0) | 141 (26.1) | 80 (14.8) | 15 (2.8) | 7 (1.3) | 2 (0.4) | 0 (0.0) | | 2 (1, 3) | |
| **Sex** | | | | | | | | | | <0.001 | | <0.001 ᴷ |
| Female | 2,197 | 563 (25.6) | 576 (26.2) | 562 (25.6) | 268 (12.2) | 143 (6.5) | 54 (2.5) | 25 (1.1) | 6 (0.3) | | 2 (1, 3) | |
| Male | 2,087 | 645 (30.9) | 649 (31.1) | 450 (21.6) | 224 (10.7) | 76 (3.6) | 35 (1.7) | 5 (0.2) | 3 (0.1) | | 2 (1, 3) | |
| Not stated | 11 | 5 (45.5) | 2 (18.2) | 3 (27.3) | 1 (9.1) | 0 (0.0) | 0 (0.0) | 0 (0.0) | 0 (0.0) | | 2 (1, 3) | |
| **Have an existing medical condition** | | | | | | | | | | <0.001 | | 0.396 ᵂ |
| No | 4,002 | 1,151 (28.8) | 1,130 (28.2) | 953 (23.8) | 445 (11.1) | 206 (5.1) | 86 (2.1) | 23 (0.6) | 8 (0.2) | | 2 (1, 3) | |
| Yes | 293 | 62 (21.2) | 97 (33.1) | 62 (21.2) | 48 (16.4) | 13 (4.4) | 3 (1.0) | 7 (2.4) | 1 (0.3) | | 2 (2, 3) | |
| **Vaccine type** | | | | | | | | | | <0.001 | | <0.001 ᴷ |
| AstraZeneca | 3,590 | 841 (23.4) | 1,007 (28.1) | 941 (26.2) | 465 (13.0) | 216 (6.0) | 85 (2.4) | 26 (0.7) | 9 (0.3) | | 2 (2, 3) | |
| J&J | 63 | 22 (34.9) | 24 (38.1) | 8 (12.7) | 7 (11.1) | 0 (0.0) | 1 (1.6) | 1 (1.6) | 0 (0.0) | | 2 (1, 3) | |
| Moderna | 61 | 20 (32.8) | 21 (34.4) | 15 (24.6) | 2 (3.3) | 0 (0.0) | 1 (1.6) | 2 (3.3) | 0 (0.0) | | 2 (1, 3) | |
| Pfizer | 57 | 26 (45.6) | 19 (33.3) | 9 (15.8) | 2 (3.5) | 0 (0.0) | 0 (0.0) | 1 (1.8) | 0 (0.0) | | 2 (1, 2) | |
| Sputnik V | 524 | 304 (58.0) | 156 (29.8) | 42 (8.0) | 17 (3.2) | 3 (0.6) | 2 (0.4) | 0 (0.0) | 0 (0.0) | | 1 (1, 2) | |
| **Dose vaccine** | | | | | | | | | | <0.001 | | <0.001 ᴷ |
| 1ˢᵗ dosage | 3,962 | 1,089 (27.5) | 1,118 (28.2) | 950 (24.0) | 470 (11.9) | 211 (5.3) | 87 (2.2) | 28 (0.7) | 9 (0.2) | | 2 (1, 3) | |
| 2ⁿᵈ dosage | 213 | 80 (37.6) | 74 (34.7) | 37 (17.4) | 16 (7.5) | 5 (2.3) | 1 (0.5) | 0 (0.0) | 0 (0.0) | | 2 (1, 3) | |
| Not specified | 120 | 44 (36.7) | 35 (29.2) | 28 (23.3) | 7 (5.8) | 3 (2.5) | 1 (0.8) | 2 (1.7) | 0 (0.0) | | 2 (1, 3) | |

n (%ʳ): Frequency (row percentage). K: P-value from the Kruskal Wallis test of equal median across groups.

W: Wilcoxon rank sum test of test of equality of median between 2 groups. All other p-values are from the Fishers exact test.

related to the AstraZeneca vaccine and 1 incident of cerebrovascular accident also associated with AstraZeneca. For the Moderna vaccine, there was 1 case of gastroenteritis and 1 case of Stevens-Johnson syndrome. Seven of these events occurred after the first dose except for Stevens-Johnson syndrome for which the dose number was not stated.

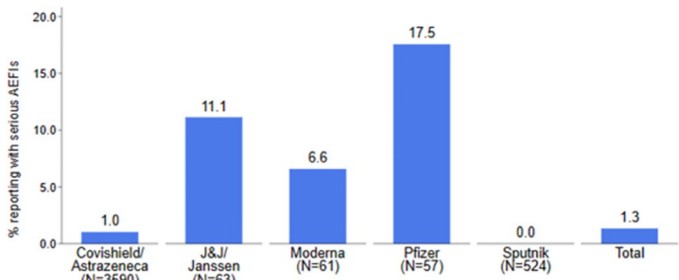

**Fig 2. Incidence of serious AEFIs by vaccine type in Ghana.**

## Outcome of AEFIs among those who reported AEFIs

Among the 4,295 persons who experienced AEFIs, 54.1% were still recovering, 42.1% recovered, 3.3% was not indicated and 0.4% were documented as deaths (Fig 4).

Table 6 below shows the distribution of the outcomes of AEFIs across the reported characteristics. The bivariate analysis showed a significant association between the outcome of the

**Table 5. Factors associated with serious AEFIs experienced.**

| Characteristics | Serious AEFIs | | Unadjusted logistic regression | | Adjusted logistic regression | |
|---|---|---|---|---|---|---|
| | n/N (%) | P-value | cOR [95% CI] | P-value | aOR [95% CI] | P-value |
| **Overall** | 57/4,295 (1.3) | | | | | |
| **Age group** | | 0.002 [χ] | | 0.004 [#] | | 0.009 [#] |
| <30 years | 15/1,317 (1.1) | | 0.38 [0.19, 0.77] | 0.007 | 0.40 [0.19, 0.86] | 0.019 |
| 30–49 years | 22/1,860 (1.2) | | 0.39 [0.21, 0.75] | 0.004 | 0.52 [0.26, 1.05] | 0.070 |
| 50+ years | 17/578 (2.9) | | 1.00 [reference] | | 1.00 [reference] | |
| Not specified | 3/540 (0.6) | | 0.48 [0.14, 1.68] | 0.254 | 0.36 [0.11, 1.19] | 0.094 |
| **Sex** | | 0.041 [χ] | | | | |
| Female | 33/2,197 (1.5) | | 1.00 [reference] | | 1.00 [reference] | |
| Male | 23/2,087 (1.1) | | 0.73 [0.43, 1.25] | 0.251 | 0.79 [0.45, 1.38] | 0.404 |
| Not stated | 1/11 (9.1) | | 6.56 [0.82, 52.72] | 0.077 | 30.27 [4.55, 201.39] | <0.001 |
| **Have existing medical condition** | | 0.560 [χ] | | | | |
| No | 52/4,002 (1.3) | | 1.00 [reference] | | 1.00 [reference] | |
| Yes | 5/293 (1.7) | | 1.32 [0.52, 3.33] | 0.558 | 1.66 [0.63, 4.36] | 0.301 |
| **Vaccine type** | | <0.001 [χ] | | <0.001 [#] | | <0.001 [#] |
| J&J | 7/63 (11.1) | | 1.00 [reference] | | 1.00 [reference] | |
| AstraZeneca | 36/3,590 (1.0) | | 0.08 [0.03, 0.18] | <0.001 | 0.05 [0.02, 0.12] | <0.001 |
| Moderna | 4/61 (6.6) | | 0.59 [0.17, 2.01] | 0.398 | 0.15 [0.03, 0.63] | 0.010 |
| Pfizer | 10/57 (17.5) | | 1.67 [0.60, 4.59] | 0.324 | 0.32 [0.09, 1.16] | 0.083 |
| Sputnik V | 0/524 (0.0) | | 0.01 [0.00, 0.13] | 0.001 | <0.01 [<0.01, 0.07] | <0.001 |
| **Dose vaccine** | | <0.001 | | | | |
| 1st dosage | 38/3,962 (1.0) | | 1.00 [reference] | | 1.00 [reference] | |
| 2nd dosage | 2/213 (0.9) | | 0.98 [0.23, 4.09] | 0.977 | 0.99 [0.27, 3.69] | 0.991 |
| Not specified | 17/120 (14.2) | | 17.04 [9.31, 31.20] | <0.001 | 10.96 [4.72, 25.46] | <0.001 |

n/N (%): Frequency/Total (percentage). cOR: crude odds ratio. aOR: adjusted odds ratio. CI: confidence interval.

[χ]: P-value from Pearson chi-square test. All other p-values are from the binary logistic regression model.

[#]: Test of overall significance of variable

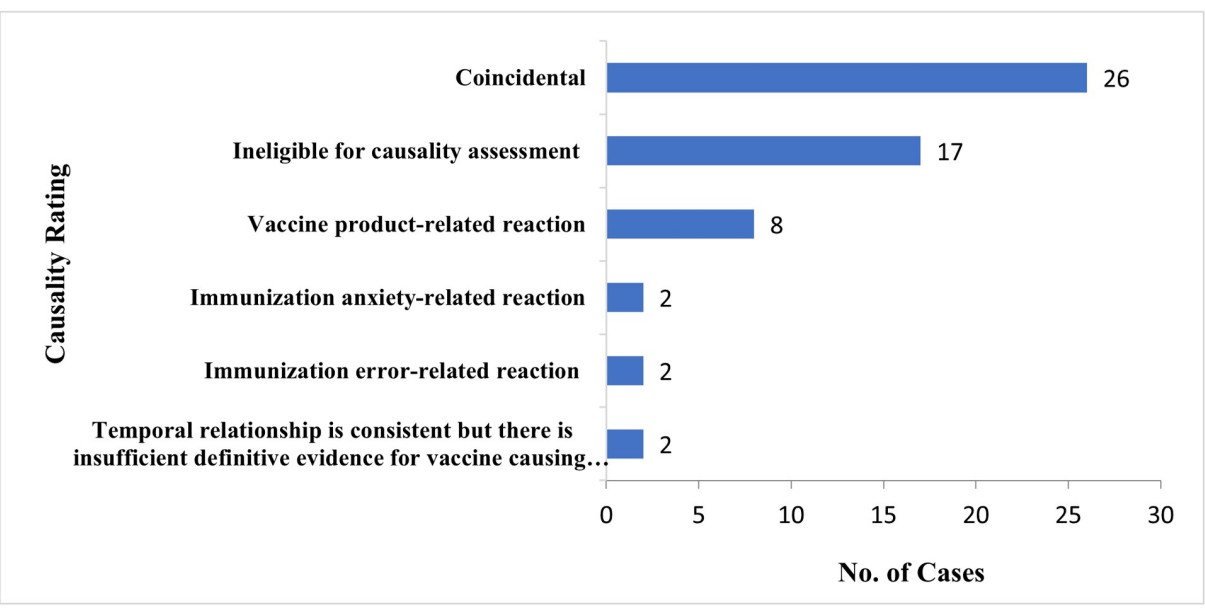

**Fig 3. Causality assessment of all serious AEFIs in Ghana.**

AEFIs and all the observed characteristics (p<0.001). The median age of those documented deaths was 51.5 years (IQR: 28–63) which was higher compared to those with other outcomes (Table 6).

## Discussion

Overall, the cumulative incidence of AEFIs among those vaccinated as of 30th June 2022 in the spontaneous reports was 24.7 per 100,000 persons.

The highest AEFI incidence among the 5 vaccines administered nationwide was observed among those who received the Sputnik V vaccine whereas Pfizer recorded the lowest. This finding corroborates a study by Kant et al., [35] which also showed a low incidence of AEFIs to Pfizer vaccine compared with persons who received AstraZeneca, Moderna and Janssen COVID-19 vaccines in The Netherlands. In contrast to the high incidence of AEFIs from the Sputnik V vaccine in Ghana, Hasan et al., [36] reported a rather low incidence of AEFIs following the administration of the vaccine in Pakistan. The report however attributed the lower

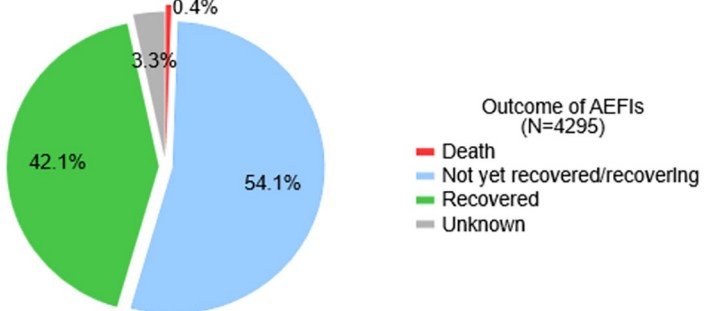

**Fig 4. Outcome of AEFIs among the vaccine recipients in Ghana.**

**Table 6. Outcome of AEFIs.**

| Characteristics | Total | Outcome of AEFIs | | | | P-value |
|---|---|---|---|---|---|---|
| | | Death | Not yet recovered | Recovered | Unknown | |
| | N = 4,295 | N = 19 | N = 2,324 | N = 1,809 | N = 143 | |
| | N | n (%ʳ) | n (%ʳ) | n (%ʳ) | n (%ʳ) | |
| Age, Median (IQR) | 33 (28–41) | 51.5 (28–63) | 33 (28–42) | 32 (27–39) | 33 (27–43) | <0.001 [k] |
| Age group | | | | | | <0.001 |
| <30 years | 1,317 | 5 (0.4) | 684 (51.9) | 586 (44.5) | 42 (3.2) | |
| 30–49 years | 1,860 | 4 (0.2) | 988 (53.1) | 824 (44.3) | 44 (2.4) | |
| 50+ years | 578 | 9 (1.6) | 347 (60.0) | 201 (34.8) | 21 (3.6) | |
| Not specified | 540 | 1 (0.2) | 305 (56.5) | 198 (36.7) | 36 (6.7) | |
| Sex | | | | | | <0.001 |
| Female | 2,197 | 9 (0.4) | 1,183 (53.8) | 922 (42.0) | 83 (3.8) | |
| Male | 2,087 | 9 (0.4) | 1,138 (54.5) | 880 (42.2) | 60 (2.9) | |
| Not stated | 11 | 1 (9.1) | 3 (27.3) | 7 (63.6) | 0 (0.0) | |
| Have an existing medical condition | | | | | | <0.001 |
| No | 4,002 | 17 (0.4) | 2,104 (52.6) | 1,748 (43.7) | 133 (3.3) | |
| Yes | 293 | 2 (0.7) | 220 (75.1) | 61 (20.8) | 10 (3.4) | |
| Vaccine type | | | | | | <0.001 |
| AstraZeneca | 3,590 | 14 (0.4) | 2,194 (61.1) | 1,286 (35.8) | 96 (2.7) | |
| J&J | 63 | 1 (1.6) | 36 (57.1) | 21 (33.3) | 5 (7.9) | |
| Moderna | 61 | 1 (1.6) | 21 (34.4) | 1 (1.6) | 38 (62.3) | |
| Pfizer | 57 | 3 (5.3) | 42 (73.7) | 8 (14.0) | 4 (7.0) | |
| Sputnik V | 524 | 0 (0.0) | 31 (5.9) | 493 (94.1) | 0 (0.0) | |
| Dose vaccine | | | | | | <0.001 |
| 1ˢᵗ dosage | 3,962 | 14 (0.4) | 2,147 (54.2) | 1,687 (42.6) | 114 (2.9) | |
| 2ⁿᵈ dosage | 213 | 1 (0.5) | 127 (59.6) | 71 (33.3) | 14 (6.6) | |
| Not specified | 120 | 4 (3.3) | 50 (41.7) | 51 (42.5) | 15 (12.5) | |
| Seriousness of AEFIs | | | | | | <0.001 |
| Not serious | 4,238 | 3 (0.1) | 2,302 (54.3) | 1,792 (42.3) | 141 (3.3) | |
| Serious | 57 | 16 (28.1) | 22 (38.6) | 17 (29.8) | 2 (3.5) | |

n (%ʳ): Frequency (row percentage). IQR: interquartile range

[k]:P-value from the Kruskal Wallis test. All other p-values are from the Pearson chi-square test.

incidence of AEFIs in the Pakistani population to underreporting due to a less robust pharmacovigilance system. It is worth mentioning that because the reports were received from the spontaneous reporting pathway, this observation may not be a true reflection of what pertains to Ghana due to the general underreporting of AEFIs in Ghana.

The most common AEFIs recorded across the 5 COVID-19 vaccines were headache, pain, pyrexia, chills, and injection site pain. These events were usually mild to moderate. This is similar to what was observed during the early developmental phase and the real-world administration of these vaccines [35, 36].

Overall, out of the 4,295 persons, at least one incidence of AEFI occurred among persons aged between 20–49. This observation is consistent with most studies involving all 5 vaccines, indicating that younger age groups are more susceptible to higher AEFI incidences [37–39].

With sex, the overall incidence of AEFIs was significantly higher among females than in males. This higher incidence among females was observed across AstraZeneca, Moderna and

Pfizer which is consistent with reports from previous studies [30, 40–42]. However, for the Sputnik V and the J&J vaccines, more males reported a higher incidence of AEFIs. Although this may not be the norm from literature, Hasan et al., [36] reported that generally there was a higher incidence of AEFIs in men than in women in Pakistan when they began their mass vaccination. The vaccines rolled out in Pakistan included all five vaccines used in Ghana and overall, more men than women had been fully vaccinated at the time of the report, hence accounting for the higher incidence in males. In Ghana, however, as of 30th June 2022, more females than males (about 54.2%) had received at least one dose of a vaccine [26].

In the current study, regression analysis showed that the incidence of serious AEFI was independent of sex. This observation was consistent with the study by Odeigah et al., [43] although these contradicted reports found in other literature [30, 38].

Apart from the J&J COVID-19 vaccine which is given as a single dose in the primary vaccination series, all the other 4 vaccines are given as a 2-dose series in the primary vaccination. More AEFIs were reported by vaccine recipients after the 1st dose (92.2%) than after the 2nd dose for all the 4 vaccine types used in Ghana. This observation is consistent with reports from early phase trials and real-life administration studies on the AstraZeneca vaccine [44–46]. For the Moderna vaccine, however, contrary to what was observed in the Ghanaian study, reports from other studies [47, 48] indicate an increase in AEFI incidence after dose 2. Similarly for the Pfizer vaccine (also an mRNA vaccine like the Moderna vaccine), studies indicate an increase in AEFIs after the second dose [42, 49]. Mathioudakis et al., [50] indicated that this increase in reactogenicity of mRNA vaccines is even more pronounced after a second dose when vaccine recipients have had a prior COVID-19 infection. Unfortunately, information on prior COVID-19 infections among the Ghanaian vaccine recipients was not obtained at the time of reporting; hence, the observed disparity cannot be properly assessed.

About 6.8% of those who reported AEFIs had a history of a chronic medical condition with hypertension, allergies, ulcers, and diabetes as the most common among them. However, in the adjusted logistic regression model, an existing chronic medical condition was not associated with the seriousness of the AEFI experienced. Nevertheless, although it is highly recommended to vaccinate these high-risk populations, it is important to closely monitor them. This is because most of the developmental phase studies of the vaccines did not involve such vulnerable populations; hence there is a degree of significant research gaps in this area which may require further studies. The recommendations for vaccinating these special populations against COVID-19 will continue to evolve as research advances [51].

From the Adjusted Logistic Regression model of factors associated with the seriousness of AEFIs among people who reported COVID-19 vaccine-related AEFIs, relative to those 50 years and above, the odds of serious AEFI was 60% less among those aged <30 years (aOR = 0.40, CI: [0.19, 0.86], p = 0.019).

It is, however, worth noting that of the 57 serious AEFIs, the causality assessment by the FDA's Joint COVID-19 Vaccine Safety Review Committee indicated that only 8 (14%) were vaccine product-related reactions providing further assurance of the safety of the vaccines in Ghana.

The eight serious AEFIs which resulted in hospitalization were assessed by the FDA's Joint COVID-19 Vaccine Safety Review Committee using the WHO Guidelines on Causality Assessment of AEFIs as having a Consistent causal association to immunization (A1: Vaccine product-related) [33]. As indicated under the results section, of the eight cases, five and one were diagnosed as febrile illness and cerebrovascular accident respectively with the AstraZeneca vaccine. The remaining two cases were gastroenteritis and Stevens-Johnson syndrome for the Moderna vaccine. Seven of these events occurred after the first dose except for Stevens-Johnson syndrome for which the dose number was not stated. The eight vaccine recipients

who had serious AEFIs fully recovered. None of these serious AEFIs were considered new because they were reported in the summary of product characteristics for the two products [52]. The strengths of this study on spontaneous COVID-19 vaccine safety reports include the study population from across different parts of the country, the inclusion of various variables, like the history of chronic medical conditions, vaccine doses received and the availability of different vaccine brands in Ghana which allowed for comparison. This study is limited by its use of a cross-sectional design that relied on spontaneously reported outcomes/symptoms, a situation which may sometimes be subjective (information bias) and may not depict the real adverse events experienced.

## Conclusion

This study provides evidence on the COVID-19 vaccine-related AEFIs which were reported through the spontaneous reporting pathway following their deployment in Ghana during the mass administration campaign. In general, the incidence of AEFI was low with an even lower incidence of serious AEFIs. The cumulative incidence of AEFIs across the 5 vaccines was about 25 per 100,000 vaccinated persons. Relative to those 50 years and above, the odds of serious AEFI was 60% less among those aged <30 years. However, a causality assessment indicated that few serious AEFIs were rated vaccine product-related reactions. Informing the public about the safety of the vaccines as well as the potential side effects, may increase public trust and acceptance decreasing vaccine hesitancy in current and future vaccination programmes. Future research focused on assessing AEFIs from active surveillance will be needed to ascertain the low incidence of serious AEFIs among the Ghanaian population.

## Acknowledgments

This study was supported by the Ghana Food and Drugs Authority (GFDA). The authors acknowledge the Management of the GFDA, the Ghana Health Service Expanded Programme on Immunization, and the Government of Ghana.

## Author Contributions

**Conceptualization:** Amma Frempomaa Asare, Harriet Affran Bonful.

**Data curation:** Amma Frempomaa Asare, Yakubu Alhassan.

**Formal analysis:** Amma Frempomaa Asare, Yakubu Alhassan.

**Methodology:** Amma Frempomaa Asare, Yakubu Alhassan.

**Writing – original draft:** Amma Frempomaa Asare, Yakubu Alhassan, Harriet Affran Bonful.

**Writing – review & editing:** George Tsey Sabblah, Richard Osei Buabeng, Abena Asamoa-Amoakohene, Kwame Amponsa-Achiano, Naziru Tanko Mohammed, Delese Mimi Darko.

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
