## [Decision Letter · Decision Letter 0]

9 Jul 2024

PGPH-D-24-01195

Adverse events following COVID-19 vaccination: A comprehensive analysis of spontaneous reporting data in Ghana

Dear Dr. Bonful,

Thank you for submitting your manuscript to PLOS Global Public Health. After careful consideration, we feel that it has merit but does not fully meet PLOS Global Public Health’s publication criteria as it currently stands. Therefore, we invite you to submit a revised version of the manuscript that addresses the points raised during the review process.

Reviewer 1 in particular had important suggestions you should consider.

We look forward to receiving your revised manuscript.

Kind regards,

Abram L. Wagner, PhD, MPH

Academic Editor

Journal Requirements:

1. Please send a completed 'Competing Interests' statement, including any COIs declared by your co-authors. If you have no competing interests to declare, please state "The authors have declared that no competing interests exist".

a. Please clarify all sources of funding (financial or material support) for your study. List the grants (with grant number) or organizations (with url) that supported your study, including funding received from your institution. 

b. State the initials, alongside each funding source, of each author to receive each grant.

c. State what role the funders took in the study. If the funders had no role in your study, please state: “The funders had no role in study design, data collection and analysis, decision to publish, or preparation of the manuscript.”

If you did not receive any funding for this study, please simply state: “The authors received no specific funding for this work.”"

Additional Editor Comments (if provided):

Reviewers' comments:

Reviewer's Responses to Questions

**Comments to the Author**

1. Does this manuscript meet PLOS Global Public Health’s publication criteria? Is the manuscript technically sound, and do the data support the conclusions? The manuscript must describe methodologically and ethically rigorous research with conclusions that are appropriately drawn based on the data presented.

Reviewer #1: Yes

Reviewer #2: Yes

Reviewer #3: Yes

2. Has the statistical analysis been performed appropriately and rigorously?

Reviewer #1: Yes

Reviewer #2: Yes

Reviewer #3: I don't know

3. Have the authors made all data underlying the findings in their manuscript fully available (please refer to the Data Availability Statement at the start of the manuscript PDF file)?

Reviewer #1: Yes

Reviewer #2: Yes

Reviewer #3: Yes

4. Is the manuscript presented in an intelligible fashion and written in standard English?

Reviewer #1: Yes

Reviewer #2: Yes

Reviewer #3: Yes

5. Review Comments to the Author

Reviewer #1: Thank you for the opportunity given me to review this paper which was well written.

Introduction

- Line 56: The authors may consider revising “method of injection” to “method of administration”. Since one is still in the introduction section and speaking in general terms, it may be wise to not limit yourself to a single mode of vaccine administration as different modes of administration will have different impacts on certain individuals.

- Authors refer to the deployment of a mass COVID-19 vaccination programme. Consider adding the date the mass vaccination programme kicked off in Ghana.

- You may also want to consider adding some context around when each of the various vaccines were introduced during this mass vaccination programme or were they all granted emergency use authorization at the same time in the country?

Study Design

- It may be useful to explain the rational for choosing the cut off dates of 2nd March 2021 to 30th June 2022 for the data used in this study. For example, was the 2nd of March the date the 1st spontaneous report was received? Or was this period the period of the mass vaccination programme?

Study variables

- Data collection and handling processing

o Line 166: Please add the version of MedDRA that was used.

Results

- Line 199: The authors may want to rephrase this sentence to make it clearer; firstly, when you talk about most reports, which reports specifically are you referring to (the total of 8498 or only the spontaneous reports)? You also outline most reports coming from the 5 different vaccines and it’s a bit difficult to understand the 83.6%. And lastly if most reports came from the aforementioned 5 vaccines, where did the other reports come from?

- Table 1:

o The authors may want to consider adding a footnote explaining the 0% of dosage 2 for the J&J vaccine to remind readers that this vaccine requires only 1 dose for a primary vaccination series.

o It may be easier to read if the totals of the medical conditions are arranged in order of most common to least common with other medical conditions last. A foot note describing what constituted “other medical conditions” will also be helpful.

- Table 2: The authors may want to consider re-arranging the total types of AEFIs in order of most to least common with Other AEFIs last.

Serious AEFIs

- A major thing missing from this section is a detailed description of the diagnoses of the serious AEFIs reported, especially the 8 that were deemed to be causally related to the vaccines. One suggestion would be to describe them by vaccine type, dose, seriousness criteria (deaths, hospitalizations etc) and outcome. The justification for this is that even though most AEFIs are mild, it’s the very few serious ones that contribute more to vaccine hesitancy and require more targeted public health messaging. So having more information on these will be important.

Discussion

- Line 298-301: The authors attribute the low incidence of AEFIs amongst people that received the Pfizer vaccine to the low deployment of this vaccine in Ghana. However, from table 3, Pfizer was the 2nd highest vaccine administered. The data presented and the conclusion drawn from it do not seem to correspond.

- The discussion section is silent on how the serious AEFIs (especially those considered vaccine-related) compared with other studies.

- In addition to the cross-sectional design, could the fact that only spontaneous reports from passive surveillance be also a reasonable limitation to this study? Active surveillance contributed to about half of all the reports received. Could these AEFIs differ significantly from the spontaneous ones as spontaneous AEFIs are notoriously underreported?

Reviewer #2: Thank you for the opportunity to review this manuscript titled “Adverse Events Following COVID-19 Vaccination: A Comprehensive Analysis of Spontaneous Reporting Data in Ghana.” This topic is a good area to research considering the short period in producing the COVID-19 vaccines, the lack of information about the clinical trials, and the misconceptions people had and continue to have about the vaccines.

Kindly consider the following review to strengthen your paper:

Abstract

Well-stated abstract. It gives a brief overview of the entire study.

Page 2, line 37: Please give the values for the results of the logistic regression for the statement “Factors associated with serious AEFIs were age (AOR. 95% CI. p-value) and vaccine type.”

Introduction

Well-discussed introduction from the perspective of what AEFIs is, how it was rapidly developed with few long-term follow-ups on the clinical trials, reported cases globally (prevalence), the commonly reported reactions in the context of the specific vaccines as well as the context-specific for Ghana, and the need for this study.

To further strengthen your introduction, please provide a brief description of how the reported adverse drug reactions were managed, either by local means, hospitalization, or self-limiting.

Materials and Methods

Page 6, line 132: Changed “employs” after “assessment” to “employed.”

Page 9, line 199: The number for AstraZeneca is omitted. Since the others were quoted, it is good you quote that too.

Page 13, line 227: Delete “the” before “those.”

Discussion

Well-discussed results with supporting references.

Conclusion

It is very succinct and reflective of the purpose of the study.

Reviewer #3: This is a good manuscript. I understand that the first vaccine safety trials were done elsewhere note in Ghana, kindly consider to capture data for the local context if available. So that you can have a true reflection of AEFIs based in Ghana

6. PLOS authors have the option to publish the peer review history of their article (what does this mean?). If published, this will include your full peer review and any attached files.

**Do you want your identity to be public for this peer review?** For information about this choice, including consent withdrawal, please see our Privacy Policy.

Reviewer #1: **Yes: **Grace Mambula

Reviewer #2: No

Reviewer #3: No

---

## [Editor Report · Decision Letter 1]

6 Sep 2024

Adverse events following COVID-19 vaccination: A comprehensive analysis of spontaneous reporting data in Ghana

PGPH-D-24-01195R1

Dear Dr Bonful,

We are pleased to inform you that your manuscript 'Adverse events following COVID-19 vaccination: A comprehensive analysis of spontaneous reporting data in Ghana' has been provisionally accepted for publication in PLOS Global Public Health.

Best regards,

Abram L. Wagner, PhD, MPH

Academic Editor